# SNP genotypes in CYP2C9 and VKORC1 genes do not affect prostate cancer or cancer mortality among warfarin users in Finnish prostate cancer patients

Teemu J. Murtola[1,2,☉,*], Kaisa M. Skantsi[1,☉], Aino Siltari[1,3], Kirsi Talala[4], Kimmo Taari[5], Teuvo L.J. Tammela[1,2], Johanna Schleutker[6,7,8], Anssi Auvinen[9], Csilla Sipeky[6,10‡]

**1** Faculty of Medicine and Health Technology, Tampere University, Tampere, Finland, **2** TAYS Cancer Center, Department of Urology, Tampere, Finland, **3** Department of Pharmacology, Faculty of Medicine, University of Helsinki, Helsinki, Finland, **4** Finnish Cancer Registry, Cancer Society of Finland, Helsinki, Finland, **5** Department of Urology, Faculty of Medicine, University of Helsinki and Helsinki University Hospital, Helsinki, Finland, **6** Institute of Biomedicine, University of Turku, Turku, Finland, **7** FICAN West Cancer Centre, University of Turku and Turku University Hospital, Turku, Finland, **8** Department of Genomics, Laboratory Division, Turku University Hospital, Turku, Finland, **9** Prostate Cancer Research Center, Unit of Health Sciences, Faculty of Social Sciences, Tampere University, Tampere, Finland, **10** UCB Biopharma, Translational Medicine, Braine l'Alleud, Belgium

☉ These authors contributed equally to this work.
‡ Contribution made in Institute of Biomedicine, University of Turku, Turku, Finland.
* teemu.murtola@tuni.fi

## Abstract

The coagulation cascade is thought to contribute to cancer progression. Although *in vitro* studies suggest that anticoagulants, such as warfarin, might reduce cancer progression, epidemiological data indicate that warfarin users may have a higher risk of cancer mortality. However, single nucleotide polymorphisms (SNPs) that influence warfarin dosing might affect this association. We investigated the risk associations between warfarin use and prostate cancer (PCa) survival, considering the SNP genotypes of *CYP2C9* and *VKORC1*, which are known to impact both warfarin pharmacokinetics and pharmacodynamics, resulting in lower warfarin dose requirement. We genotyped 2,246 Finnish men with PCa from two different cohorts for SNPs rs1057910, rs1799853, and rs9923231. Genotyping was done using a custom Illumina iSelect genotyping array (iCOGs). Using Cox regression models, we calculated hazard ratios (HRs) and 95% confidence intervals (CI) for the risk of overall death, cancer deaths overall, and PCa-specific death after PCa diagnosis based on SNP genotypes. Data on warfarin purchases was obtained from a national registry. Our findings revealed that the SNPs did not alter the risk of cancer or PCa death in either cohort, nor did they modify the risk among warfarin users. However, overall mortality was higher among warfarin users compared to non-users, particularly in carriers of all three SNPs. Even though the increased mortality is likely due to confounding by indication, warfarin use may increase overall mortality especially in men with lower

**Data availability statement:** There are legal restrictions which prevent the public sharing of minimal data for this study. The data contains sensitive personal health information which the authors are unable to share openly. Data are available upon request from the Finnish Social and Health Data Authority, Findata, via the Findata website (https://findata.fi/en/) or email (info@findata.fi), for researchers who meet the criteria for access to confidential data.

**Funding:** FinRSPC has been supported by grants from Academy of Finland (grant #260931), Cancer Foundation Finland sr, and Expert Responsibility Area of the Pirkanmaa Hospital District (grant #9V065) (A.A). This study was supported by grant from Cancer Foundation Finland (grant #220055) (T.M). The funders had no role in the study design, data collection and analysis, decision to publish, or preparation of the manuscript. There was no additional external funding received for this study by other authors of this manuscript.

**Competing interests:** The authors have declared that no competing interests exist.

warfarin dose requirements due to SNP carrier status. However, we need further studies with larger populations to confirm these findings.

## Introduction

Prostate cancer (PCa) is the second most common cancer in men and the second most common cause of cancer deaths in countries with a high Human Development Index [1]. The risk factors for the initiation and progression of PCa are still partly unknown. Established non-modifiable risk factors are age, ethnicity, and PCa family history. The factors determining and affecting prognosis, apart from clinical tumor characteristics, remain poorly characterized. Modifiable factors, such as diabetes, obesity, and hypercholesterolemia, are associated with worse prognosis in some studies, however, their role remains unclear [2–4].

The coagulation cascade has been proposed to play a role in the hematogenous spread of cancer and the formation of metastases. Platelets adhere to circulating cancer cells, protecting them from the immune system [5–7]. Platelet RNA is altered in cancer patients, and platelets could serve as a potential cancer biomarker [8]. Therefore, it would be logical to assume that anticoagulants, which affect platelet function, would decrease metastatic spread by reducing platelet protection of circulating cancer cells.

Previous preclinical studies have reported that the widely used anticoagulant warfarin may protect against tumor cell spreading by a mechanism related to modulation of the coagulation cascade rather than a direct impact on cancer cells [9–11]. For PCa, warfarin use has been associated with a decreased risk in one epidemiological study [12], but not all studies agree [13]. Furthermore, in epidemiological studies of cancer mortality, the risk of cancer death is increased among warfarin users compared to non-users [14–16]. The epidemiological risk association is likely affected by reverse causation, i.e., anticoagulants are used due to the well-established elevated risk of thrombosis related to advanced cancer, and not vice versa [16–22]. However, causal association cannot be ruled out entirely.

Warfarin is metabolized in the liver via the enzyme cytochrome P450 family 2 subfamily C member 9 (CYP2C9). Warfarin targets the Vitamin K epOxide Reductase Complex subunit 1 (VKORC1) by inhibiting its activity, an enzyme involved in the vitamin K cycle. Variant alleles, i.e., single nucleotide polymorphisms (SNPs) of both *CYP2C9* and *VKORC1* genes are associated with lower warfarin dose requirements [23–25]. In addition, several other genes, such as *CYP1A2* and *CYP3A4*, are also associated with warfarin dose requirements [26]. Therefore, if warfarin indeed has a direct, causal effect on cancer mortality *in vivo,* it could be logically assumed that the carrier status of SNPs in *CYP2C9* and *VKORC1* would modify the risk association between warfarin use and cancer mortality. As far as we know, this topic has not been studied before. It is also unknown whether SNP variations in these specific warfarin pharmacogenes independently associate with cancer death.

Therefore, we first explored the association of SNPs rs1057910, rs1799853, and rs9923231 in warfarin pharmacogenetics genes *CYP2C9* and *VKORC1* with cancer

mortality to evaluate their role as independent predictors of cancer death using two different cohorts. Secondly, we evaluated whether these SNPs modify the risk association of warfarin use with PCa prognosis, cancer-specific and overall death. In this way, we aim to indirectly explore whether warfarin use has a direct effect on cancer progression.

## Materials and methods

### Study populations

We used two study cohorts named the Screening cohort and the Hospital cohort. The Screening cohort included men from the Finnish Randomized Study of Screening for Prostate Cancer (FinRSPC) [27]. In FinRSPC, 80,458 men from Tampere and Helsinki regions initially aged 55−67 years were identified from the nationwide Population Register Centre and randomized for screening with prostate-specific antigen (PSA) every fourth year (screening arm) or to be followed via national registries (control arm). Information on new PCa diagnoses during 1996−2015 were obtained from the Finnish Cancer Registry [28] and complemented with information from hospital records. Data quality of Finnish register data is high. For instance, the Finnish Cancer Registry covers annually 96% of all diagnosed solid tumors [28,29]. The data includes the date of diagnosis, Gleason score, TNM classification, and PSA value. Dates and causes of death occurring during 1996−2015 were obtained from Statistics Finland's Causes of Death registry (permission no TK/3536/07.03.00/2021). Causes of death are recorded according to ICD-10 coding; deaths with PCa (ICD-10 code C61) as the primary cause of death were considered as PCa-specific deaths, and deaths with any cancer (C00-C97) as the primary cause were considered cancer-specific deaths. A data quality assessment comparing classification of causes of death in the Causes of Death registry and an independent causes of death committee observed small but statistically insignificant differential misclassification bias [30], thus, data quality from the registry is estimated to be sufficient for our analyses. For these analyses, the study population was limited to 810 men with PCa in the screening arm.

The Hospital cohort population consisted of 1,547 men diagnosed and treated for PCa at Tampere University Hospital, department of urology between 1996 and 2010. All men were followed after primary PCa treatment with yearly follow-up visits including a PSA test and clinical examination. Bone scans were performed upon suspicion of metastases at the discretion of the attending urologist. Available clinical data between 1996 and 2020 included the year of diagnosis, tumor Gleason score, TNM-stage, and PSA at diagnosis, primary treatment method, disease recurrences (biochemical or radiological) during the follow-up visits, and the date and cause of death. The data were accessed for research purposes between January 1, 2022, and October 31, 2024.

Both studies were accepted by the Ethical board of Pirkanmaa wellbeing county (approval numbers ETL95057 and R03203 for FinRSPC and the hospital cohort, respectively). All experiments were carried out following guidelines and regulations. Informed consent was obtained from all subjects.

### Information on medication use

The Screening cohort was linked to the national prescription database of the Social Insurance Institution of Finland (SII) for information on outpatient warfarin purchases between 1995 and 2015. SII reimburses costs of drug purchases as part of the national health insurance. The reimbursement is paid for all drug purchases after the yearly deductible fee of 50 euros. All Finnish citizens are entitled to reimbursement for purchases of physician-prescribed drugs. The reimbursement is most commonly deducted from the purchase price directly at the pharmacy. Each reimbursed purchase is registered by the national prescription database. In Finland, warfarin is available only through physicians' prescription, thus, comprehensively documented by the registry. Detailed information for each purchase includes the purchase date, drug-specific ATC code, amount in doses, number of tablets, and packages purchased. Based on this information we estimated annual and cumulative medication use for each separate calendar year between 1995 and 2015.

A person with any purchase of warfarin during a certain year was defined as a user on that particular year. Therefore, each calendar year with warfarin purchases was marked as a year of usage. All warfarin purchases during a certain year

were summed for the annual mg amount and then divided by the average daily defined daily dose (DDD) amount of 7.5 mg [31]. Doses per year were estimated by dividing the cumulative DDD amount of each year by the cumulative years of usage.

### Genotyping of SNPs

Blood samples from men of both study cohorts were genotyped by the PRACTICAL consortium using a custom SNP panel (custom Illumina iSelect genotyping array, iCOGs) for three SNPs associated with warfarin pharmacogenetics according to previous literature; rs1057910 and rs1799853 from *CYP2C9* located in the human chromosome 10 and rs9923231 from *VKORC1* in the human chromosome 16. rs1799853 was categorized as wild type (CC), heterozygous carrier (CT), and homozygous carrier (TT). rs1057910 was categorized as wild type (AA), heterozygous carrier, one minor allele (AC), and homozygous carrier, two minor alleles (CC). rs9923231 was categorized as wild type (GG), heterozygous carrier, one minor allele (GA), or homozygous carrier, two minor alleles (AA). *CYP2C9* was categorized according to rs1057910 and rs1799853 (S4 Table). Based on literature, there is no evidence of linkage disequilibrium between CYP2C9 SNPs rs1057910 and rs1799853 [32]. CYP2C9 SNPs rs1057910 and rs1799853 status were available for 790 and 1437 subjects in the Screening and Hospital cohorts, respectively. Information on VKORC1 SNP rs9923231 status was available for all study subjects (n = 810 and n = 1527, respectively). In our data sets only 0.9% of subjects were carriers of both SNPs in the Screening cohort and 0.07% in the Hospital cohort. Information of SNP status was not used to guide clinical practice for patients.

### Statistical analyses

We performed two separate analyses. First, we compared mortality by genotype for SNPs and their combinations. Additionally, we compared mortality between warfarin users and non-users stratified by SNPs in warfarin pharmacogenetics genes *CYP2C9* and *VKORC1*. The second analysis was limited to the Screening cohort.

A Cox regression model was used to calculate hazard ratios (HRs) and 95% confidence intervals (CI) for the risk of death overall, cardiovascular disease (CVD)-specific death, PCa-specific death, the risk for PCa progression, and the risk of cancer-specific death due to any cancer type. Analysis of PCa-progression was limited to the Hospital cohort and analyses on death due to any cancer type and CVD-specific death were limited to the Screening cohort. The time metric was months and years since prostate cancer diagnosis. The censoring events in the Screening cohort were death, emigration, or Dec. 31st, 2015, and in the Hospital cohort were death, emigration, or Dec. 31th, 2020, whichever came first.

SNP carrier status was included in the Cox regression model as a categorized explanatory variable. All analyses were adjusted by age at diagnosis. Multivariable models were adjusted by age, the year of diagnosis, tumor Gleason score, tumor T stage, and PSA at diagnosis. Separate analyses were performed for the risk of PCa-specific death, death due to any cancer, and for death due to any cause using the screening cohort and PCa-specific death, risk for PCa progression, and all-cause mortality using the Hospital cohort.

To evaluate SNPs affecting warfarin pharmacogenetics as modifiers of the association between warfarin use and cancer survival, we performed similar Cox regression analysis as described above, but with warfarin user status as the explanatory variable. In these analyses, warfarin non-users were the reference group. Analysis was stratified by genotype to estimate whether the risk association with warfarin use differs by genotype. Interaction was statistically tested using the Wald test including an interaction term between warfarin use and SNP genotype in the Cox regression model. These analyses were limited to the screening cohort and further adjusted with additional drug use (aspirin use, non-steroid anti-inflammatory drug use, statin use, antidiabetics use, and antihypertensive drug use).

Warfarin use was analyzed as a time-dependent variable to minimize immortal time bias. Usage status and cumulative medication usage were updated annually according to yearly recorded medication purchases. Non-user status was

retained until the year of the first recorded purchase and the status remained as a user thereafter to minimize bias due to selective switching to fractionated heparin use in cancer patients.

Equal distribution of clinical characteristics between SNP carriers in each cohort was tested using the Chi-square test for categorical variables and the ANOVA test for linear variables. Analyses were performed using IBM SPSS Statistics for Windows, version 24 (IBM Corp., Armonk, N.Y., USA) and Stata, version 18 (StataCorp, College Station, TX, USA).

## Results

### Population characteristics by CYP2C9 and VKORC1 genotype in two study cohorts

Homozygous genotype in the studied SNPs was extremely rare in both study cohorts (with no cases in the Screening cohort and 10 cases in the Hospital cohort). Otherwise, the distributions of wild-type and heterozygous SNP carriers were similar in the two study cohorts (Table 1). There were more deaths during the follow-up in the Hospital cohort compared to the Screening cohort: the number of deaths overall in the screening cohort was 237 (28%), of which 41 (17%) were due to PCa and 74 (31%) due to other cancers. In the Hospital cohort, there were 1066 (70%) deaths overall, of which 303 (28.2%) were from PCa. Men in the Hospital cohort had a higher proportion of tumors with Gleason score 8 or higher, stage T3–T4 tumors, and metastatic diseases. Also, the median PSA was higher at the baseline compared to the screening cohort (Table 1). Median follow-up time from PCa diagnosis until death or the end of follow-up was also longer in the Hospital cohort (Table 1). Differences in PCa clinical parameters between the cohorts can largely be attributed to the differing nature of the cohorts themselves. The Hospital cohort comprises patients with symptom-detected PCa. Consequently, this cohort includes higher proportion of aggressive and advanced cases, with fewer low-grade PCa cases. In

**Table 1. Population characteristics by *CYP2C9* (rs1057910, rs1799853) and *VKORC1* (rs9923231) genotypes in two study cohorts: 790 men from the Finnish Randomized Study of Screening for Prostate Cancer (FinRSPC) (Screening cohort) and 1,436 prostate cancer patients treated at Tampere University Hospital (Hospital cohort) who had diagnosis of prostate cancer and genotypic information of all three investigated SNPs. There were no statistical differences between genotype groups in any of the variables. \*None of the subjects were homozygous variant for all SNPs, however, 22 subjects were homozygous carrier for the CYP2C9 SNPs, and 112 for the VKORC1 SNPs.**

|  | Screening cohort | | | Hospital cohort | | |
|---|---|---|---|---|---|---|
|  | Wild type | Heterozygous | Homozygous variant* | Wild type | Heterozygous | Homozygous variant |
| n (%) of men | 204 (25) | 586 (75) | 0 | 370 (26) | 1056 (73) | 10 (1) |
| n (%) of deaths | 66 (33) | 162 (28) | 0 | 251 (68) | 744 (71) | 4 (40) |
| n (%) of cancer deaths | 32 (16) | 79 (14) | 0 | 99 (27) | 302 (29) | 2 (20) |
| n (%) of prostate cancer deaths | 11 (1) | 27 (1) | 0 | 68 (18) | 213 (20) | 1 (10) |
| Median follow-up time, years (IQR) | 8.6 (7-11.5) | 8.3 (6.9-10.6) |  | 11 (5.5-15.8) | 10.8 (5-15.3) | 13.4 (9.6-15.2) |
| Age at diagnosis, median (IQR) | 66 (63-70) | 67 (63-68) | – | 71 (62-77) | 71 (63-77) | 66 (58-74) |
| Gleason score, n (%): |  |  |  |  |  |  |
| 6 or fewer | 134 (66) | 379 (65) | – | 155 (51) | 420 (49) | 5 (56) |
| 7 | 54 (27) | 143 (25) | – | 90 (30) | 268 (31) | 4 (44) |
| 8 or higher | 15 (7) | 59 (10) | – | 60 (20) | 175 (20) | 0 |
| T stage, n (%): |  |  | – |  |  |  |
| T1-T2 | 191 (94) | 531 (91) | – | 250 (68) | 712 (67) | 8 (80) |
| T3 | 10 (5) | 46 (8) | – | 76 (21) | 214 (20) | 2 (20) |
| T4 | 3 (2) | 6 (1) | – | 23 (6) | 61 (6) | 0 |
| M-stage, n (%): |  |  |  |  |  |  |
| M0/x | 137 (67) | 396 (68) | – | 315 (85) | 883 (84) | 8 (80) |
| M1 | 3 (2) | 13 (2) | – | 34 (9) | 104 (10) | 2 (20) |
| PSA, median (IQR) | 3.3 (1.9-5.89) | 3.06 (1.98-5.0) | – | 9.9 (6.6-20.9) | 11.3 (6.6-22.9) | 7.9 (5.6-15.7) |

contrast, the Screening cohort consists of PCa cases identified through population-level PSA screening program. Most of the screening-detected PCa cases were asymptomatic and involved low-grade tumors.

There were no significant differences in the median age by SNP carrier status in either cohort. Similarly,no differences were observed in median follow-up time, Gleason score, or T or M stage by genotype (Table 1).

### Risk of death by CYP2C9 and VKORC1 carrier status

Wild-type carriers of *CYP2C9* or *VKORC1,* either alone or in combination, were not associated with the risk of death overall, death from any cancer, f PCa-specific death, or risk of PCa progression (Table 2, Fig 1). The results were consistent across both study cohorts and were not modified by SNP carrier status, i.e., whether the polymorphism was heterozygous or homozygous (Table 2, Fig 1). Results from both age-adjusted (S1 Table) and multivariable-adjusted (Table 2, Fig 1) models were similar.

### Prevalence of CYP2C9 and VKORC1 carrier status among warfarin users and non-users

The prevalence of warfarin use did not differ by SNP genotype (rs1057910, rs1799853 in *CYP2C9* and rs9923231 *VKORC1)*; around 25% of men used warfarin during the study period, regardless of genotype (Table 3).

### Warfarin use as risk factor for cancer death by CYP2C9 and VKORC1 SNP carrier status

Warfarin use was associated with increased overall mortality when SNP carrier status was not considered (HR 1.76, 95% CI 1.24–2.49) (Table 4, Fig 2). However, results were inconclusive when subjects were stratified separately based on CYP2C9 and VKORC1 carrier status (Table 4, Fig 2). For instance, heterozygous carrier status of rs9923231 was associated with increased mortality among warfarin users (HR 2.35, 95% CI 1.38–4.0), while wild-type or homozygous carrier

**Table 2. Risk of overall death, cancer death, and prostate cancer (PCa)-specific death among carriers of warfarin dosing-related SNPs in two study cohorts: CYP2C9 SNPs rs1057910 and rs1799853 status were available for 790 and 1437 subjects in the Screening and Hospital cohorts, respectively. Information of VKORC1 SNP rs9923231 status was available for all study subjects (n = 810 (Screening cohort) and n = 1527 (Hospital cohort)). SNP status in analysis where all SNPs were combined included subjects with any SNPs (hetero- or homozygous) in investigated genes. Ref = reference.**

| | n of deaths/ subjects | Screening cohort | | | n of deaths/ subjects | Hospital cohort | | |
|---|---|---|---|---|---|---|---|---|
| | | Overall death | Cancer death | PCa-specific death | | Overall death | PCa progression | PCa-specific death |
| | | HR (95% CI) | HR (95% CI) | HR (95% CI) | | HR (95% CI) | HR (95% CI) | HR (95% CI) |
| *CYP2C9 (rs1057910, rs1799853)* | | | | | | | | |
| Wild-type | 154/537 | Ref | Ref | Ref | 678/981 | Ref | Ref | Ref |
| Combined heterozygous carrier | 70/231 | 1.25 (0.93-1.67) | 1.02 (0.66-1.59) | 1.5 (0.69-3.27) | 293/415 | 1.02 (0.87-1.2) | 1.2 (0.95-1.51) | 1.13 (0.83-1.54) |
| Combined homozygous carriers | 4/22 | 0.72 (0.27-1.96) | 0.77 (0.19-3.17) | 1.96 (0.25-15.14) | 29/41 | 0.98 (0.63-1.52) | 1.22 (0.68-2.18) | 0.88 (0.36-2.16) |
| *VKORC1 (rs9923231)* | | | | | | | | |
| Wild-type (GG) | 110/324 | Ref | Ref | Ref | 406/578 | Ref | Ref | Ref |
| AG/GA | 96/374 | 0.78 (0.59-1.04) | 0.86 (0.57-1.29) | 0.6 (0.28-1.27) | 498/729 | 0.93 (0.8-1.08) | 0.92 (0.74-1.15) | 1.03 (0.77-1.38) |
| AA | 31/112 | 0.83 (0.55-1.25) | 0.81 (0.44-1.50) | 1.05 (0.4-2.71) | 161/220 | 1.08 (0.88-1.33) | 0.91 (0.67-1.24) | 0.91 (0.59-1.39) |
| *All SNPs combined* | | | | | | | | |
| Wild-type | 66/204 | Ref | Ref | Ref | 251/370 | Ref | Ref | Ref |
| One or two minor alleles in both genes | 162/586 | 0.95 (0.7-1.28) | 1.02 (0.65-1.59) | 1.06 (0.47-2.4) | 744/1,056 | 1.04 (0.88-1.22) | 1.01 (0.8-1.28) | 1.12 (0.81-1.55) |

 

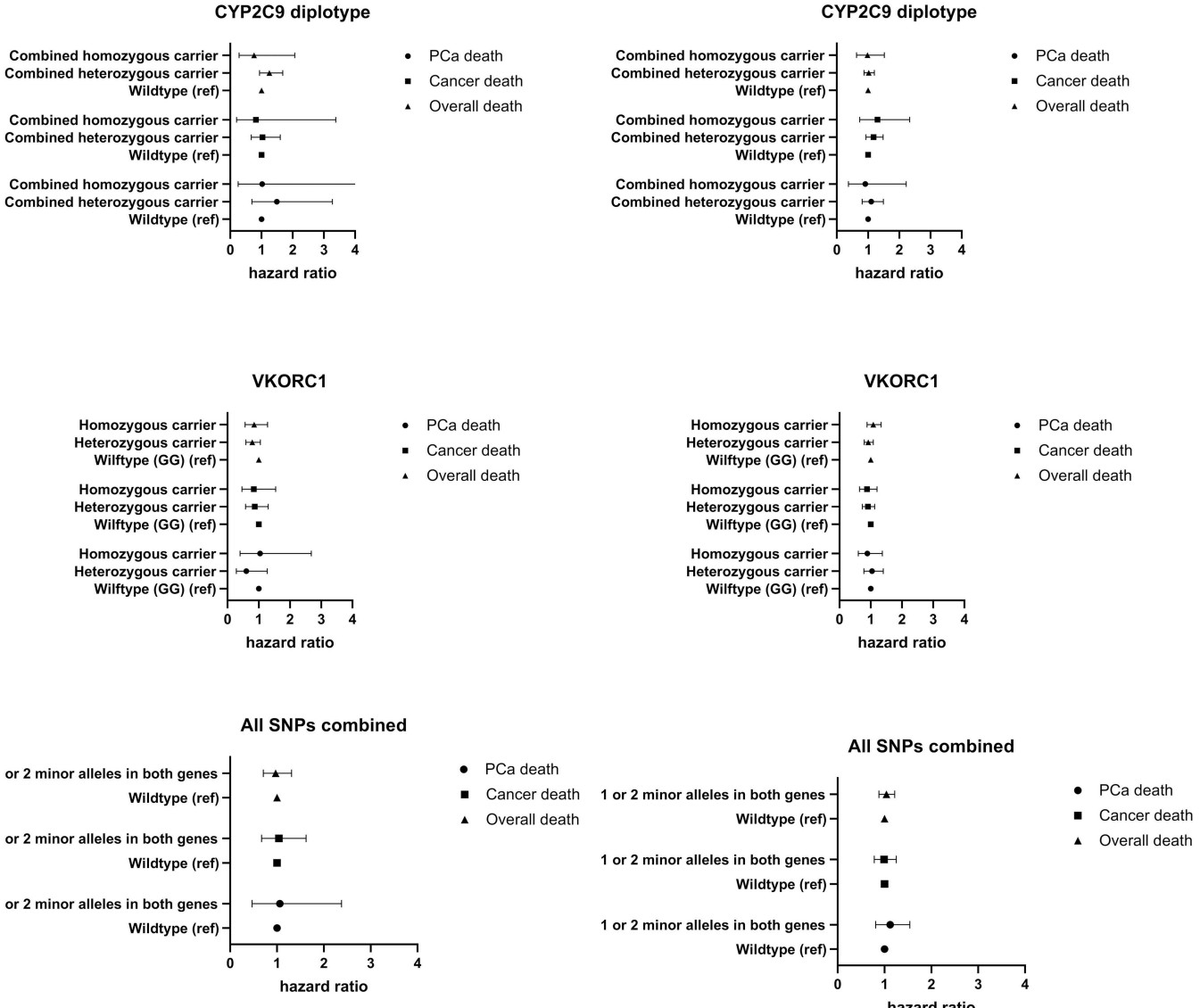

**Fig 1. Risk of prostate cancer (PCa)-specific death, cancer death, and overall death among carriers of warfarin dosing-related SNPs in two study cohorts, the Finnish Randomized Study of Screening for Prostate Cancer (Screening cohort) and the Tampere University Hospital cohort (Hospital cohort).** Data are presented as hazard ratios (HR) with 95% confidence intervals (CI).

status were not associated with increased mortality (HR 1.39, 95% CI 0.81–2.39 and HR 0.87, 95% CI 0.22–3.45, respectively). However, when stratified by carrier status across all three SNPs, warfarin use was associated with increased mortality among minor allele carriers compared to non-users with minor alleles (HR 2.06, 95% CI 1.35–3.15) (Table 4, Fig 2). No significant risk association was observed among wild-type carriers when warfarin users were compared to non-users (HR 1.35, 95% CI 0.68–2.7). The interaction term between SNP status and warfarin use was non-significant in all analyses (Table 4).

**Table 3. Warfarin use by *CYP2C9* and *VKORC1* genotype in a study cohort of prostate cancer patients from the Finnish Randomized Study of Screening for Prostate Cancer. CYP2C9 SNPs rs1057910 and rs1799853 status were available for 790 and VKORC1 SNP rs9923231 status for all study subjects (n = 810).**

| Locus genotype | All, n (%) | warfarin users, n (%) | warfarin non-users, n (%) | p-value |
|---|---|---|---|---|
| *CYP2C9* rs1057910 | | | | |
| AA | 702 (89) | 196 (28) | 506 (72) | |
| AC | 82 (10) | 22 (27) | 60 (73) | |
| CC | 6 (1) | 2 (33) | 4 (67) | |
| C carriers | 88 (11) | 24 (27) | 64 (73) | |
| C allele frequency | | | | 0.94 |
| *CYP2C9* rs1799853 | | | | |
| GG | 618 (78) | 173 (28) | 445 (72) | |
| GA | 163 (21) | 45 (28) | 117 (72) | |
| AA | 9 (1) | 1 (11) | 8 (89) | |
| A carriers | 172 (22) | 46 (27) | 125 (73) | |
| A allele frequency | | | | 0.43 |
| *VKORC1* rs9923231 | | | | |
| GG | 324 (40) | 87 (27) | 237 (73) | |
| GA | 374 (46) | 104 (28) | 270 (72) | |
| AA | 112 (14) | 32 (29) | 80 (71) | |
| A carriers | 483 (60) | 136 (28) | 350 (72) | |
| A allele frequency | | | | 0.93 |

**Table 4. Risk of overall death, cancer-specific death, and prostate cancer (PCa)-specific death by warfarin use in a population stratified by warfarin dosing SNP genotypes. Analysis was limited to screening cohort. Non-warfarin users were used as the reference group. Data is expressed as hazard ratios with 95% confidence intervals. - = number of participants with minor alleles was too low for analysis. SNP status in analysis where all SNPs were combined included subjects with any SNPs (hetero- or homozygous) in investigated genes. \*p < 0.05.**

| | n of deaths among warfarin users/non-users | Overall death | Cancer death | PCa-specific death |
|---|---|---|---|---|
| SNP status not taken into account | 71/166 | 1.76 (1.24-2.49)* | 0.98 (0.54-1.78) | 1.06 (0.38-2.99) |
| *CYP2C9 (rs1057910, rs1799853)* | | | | |
| Wild-type | 47/107 | 2.07 (1.34-3.2)* | 1.07 (0.52-2.18) | 1.68 (0.53-5.39) |
| Combined heterozygous carrier | 20/50 | 1.29 (0.62-2.69) | 0.69 (0.15-3.15) | – |
| Combined homozygous carriers | 2/2 | – | – | – |
| p-value for interaction | | 0.52 | 0.6 | 0.99 |
| *VKORC1 (rs9923231)* | | | | |
| Wild-type (GG) | 31/79 | 1.39 (0.81-2.39) | 0.71 (0.25-2.08) | 0.92 (0.1-8.32) |
| AG/GA | 33/63 | 2.35 (1.38-4.0)* | 1.18 (0.52-2.72) | 1.73 (0.41-7.33) |
| AA | 7/24 | 0.87 (0.22-3.45) | 0.27 (0.03-2.93) | – |
| p-value for interaction | | 0.31 | 0.43 | 0.61 |
| *All SNPs combined* | | | | |
| Wild-type | 20/46 | 1.35 (0.68-2.7) | 0.45 (0.1-2.06) | 0.77 (0.06-10.72) |
| One or two minor alleles in both | 49/113 | 2.06 (1.35-3.15)* | 1.4 (0.71-2.77) | 1.67 (0.51-5.47) |
| p-value for interaction | | 0.3 | 0.16 | 0.87 |

Mortality from PCa or any cancer was not associated with warfarin use, even when subjects were stratified by SNP carrier status (Table 4). Results from the age-adjusted (S2 Table) and multivariable-adjusted (Table 4) models were similar.

**Warfarin user status**

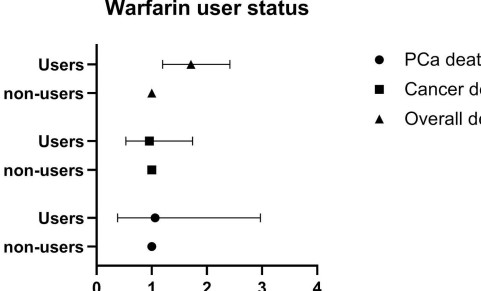

**CYP2C9 diplotype**

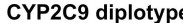
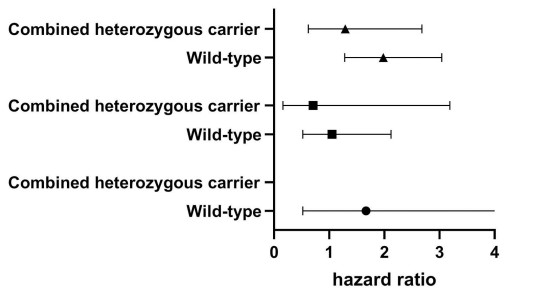

**VKORC1**

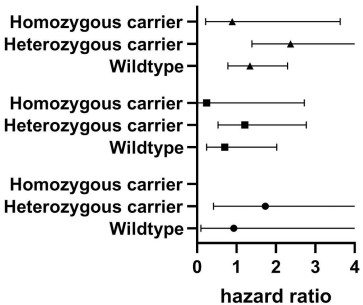

**All SNPs combined**

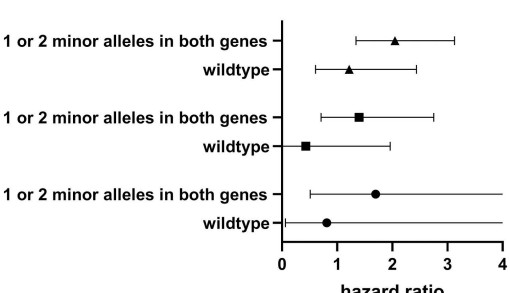

**Fig 2. Risk of prostate cancer (PCa)-specific death, cancer-specific death, and overall death by warfarin use in a population stratified by warfarin dosing SNP genotypes.** Analysis was limited to screening cohort. Non-warfarin users were used as a reference group. Data is expressed as hazard ratios with 95% confidence intervals.

Analysis of CVD-specific death showed similar trends to overall death; risk was increased among warfarin users, however, the role of SNP status remained unclear, and the interaction term between warfarin use and SNP status was also non-significant in this analysis (S3 Table).

## Discussion

The SNPs affecting warfarin dosing were not associated with clinical features of PCa, mortality from PCa, or overall cancer mortality. No association was found between warfarin use and SNP mutations. Overall and CVD-specific mortality was elevated among warfarin users compared to non-users, even when subjects were stratified by overall SNP carrier status. This elevated risk appears to be driven by non-cancer deaths, as no increase in cancer-related mortality was observed. Since warfarin pharmacogenetics genes were not associated with cancer death and did not modify the risk between warfarin use and cancer mortality, the results suggest that warfarin, or its anticoagulant effects, likely does not play a direct role in PCa progression. Thus, this indicates that the genetic factors influencing warfarin dosing do not impact the potential benefits cancer patients may receive from using warfarin.

To our knowledge, this is the first study to evaluate the association of SNPs rs1057910, rs1799853, and rs9923231 with PCa progression. However, SNP rs1057910 has previously been associated with an increased risk of developing colorectal carcinoma [33]. We did not assess whether the investigated SNPs are associated with the risk of developing PCa, therefore, this should be explored using other cohorts that include genotyping data from healthy controls as well. In PCa, VKORC1 gene expression has been studied in prostate tissue, and it was found to be higher in benign tissue compared to PCa tissue. Interestingly, SNP rs2359612, which also influences warfarin dosing, was linked to PCa risk [34]. Notably, based on a genome-wide association study (GWAS), none of the investigated SNPs are among the 451 independent variants reported to be associated with PCa risk [35].SNPs in warfarin pharmacogenetics genes, *CYP2C9* and *VKORC1,* are associated with lower warfarin dose requirements [23–25]. Theoretically, warfarin treatment in patients carrying these SNPs could lead to excessive anticoagulation, which might lead to an increased risk of bleeding. Even though we observed increased overall and CVD mortality by warfarin use, also among SNP carriers, we lack detailed information on the exact causes of death. Therefore, we cannot dretermine whether these deaths were due to warfarin-induced adverse effects, the clinical indication for warfarin use, or other factors.. However, it is important to note that these association might be influenced by confounding by indication.

Metastatic cancer is known to increase the risk of venous thrombosis through changes in the coagulation cascade [36–38]. Therefore, the elevated mortality observed among warfarin users in our study, and also in previous studies [14,39–41], is likely affected by reverse causation, i.e., thrombosis associated with advanced cancer is the indication for warfarin use. Clinically, thrombosis treatment and prevention during chemotherapy for metastatic cancer is preferably done using low molecular weight heparins (LMWHs) rather than warfarin [42]. If not properly accounted for, this could lead to bias underestimating cancer mortality among warfarin users and overestimating it among LMWH users. We aimed to minimize this bias by keeping warfarin users in the user category even if the drug purchases ceased. In such analysis, there was no protective association with warfarin use, but rather an increased risk of death.

Warfarin pharmacogenetics genotype is associated with maintenance dose requirements, accounting for up to 40–45% of individual variability [43]. Time in Therapeutic Range (TTR) should be a minimum of more than 60% to achieve the benefit of warfarin use, prevention of thromboembolic events [44]. Besides the genetic factors, International Normalized Ratio (INR) fluctuations occur due to various reasons such as INR monitoring frequency, dose adjustments, diatery vitamin K, alcohol consumption, poor adherence, malignancy, drug interactions, and congestive heart failure. Every 10% increase in warfarin dosage fluctuation is associated with a 1.58-fold increased HR of bleeding or thromboembolic events [45]. Despite optimally adjusted anticoagulation, a small proportion of patients, especially those with history of prior thromboembolism or extensive metastatic cancer, develop venous thromboembolism [46].

The SNP carriers could hypothetically be more susceptible to cancer-associated thrombosis despite warfarin use due to dose fluctuations. Nevertheless, our results do not support this hypothesis, as warfarin pharmacogenetics SNPs were not found to modify the risk associations between warfarin use and cancer outcomes.

## Limitations and strengths

The strength of our study is that we were able to combine genetic information on SNP carrier status and very detailed information on warfarin use through Finnish national registries. Further, the impact of SNP carrier status was evaluated using two population-based cohorts. Our data allowed examination of PCa progression and survival, overall cancer mortality, CVD mortality, and all-cause mortality. To our knowledge, this is the first study to examine these associations. By connecting warfarin pharmacogenetics to warfarin user status, we were able to draw conclusions of whether warfarin use affects PCa prognosis and outcome.

A limitation of our study is the relatively small sample size, which limited statistical power to detect risk associations among warfarin users, especially among homozygous SNP carriers. Also, the small number of PCa deaths limited our ability to estimate PCa-specific survival. The Finnish population is predominantly Caucasian and genetically homogenous [47–49], therefore, the generalizability of our findings to other ethnic groupss is limited and needs confirmation in more diverse populations. Even though the data about warfarin purchases are reliable and accurate, it needs to be considered that we do not know whether subjects used the purchased drugs or not. Since warfarin dosing is adjusted based on a patient's INR, it would be valuable to classify patients as good or poor responders and analyze outcomes accordingly. However, INR values were not available in this dataset, thus, these analyses need to be done using other data sets. Our results may also be influenced by residual confounding from unknown systematic differences between warfarin users and non-users. Thus, confirmation from larger, independent datasets will be needed.

## Conclusion

SNP variants in the *CYP2C9* and *VKORC1* genes, which affect warfarin dosing, do not modify the association between warfarin use and PCa mortality. This suggests that the pharmacogenetics of warfarin dosing does not affect how a cancer patient may benefit from warfarin usage. Warfarin use was associated with increased overall and CVD mortality compared to non-users, including men carrying minor *CYP2C9* and *VKORC1* alleles. These findings should be validated in larger study populations with more outcome events. Also, studies estimating cancer survival by anticoagulant use should account for potential selection bias due to cancer-related increases in thrombosis risk.

## Supporting information

**S1 Table. Results from the age-adjusted cox regression model with SNP status for overall death, cancer specific-death, prostate cancer-specific death, and prostate cancer (PCa) progression.**
(XLSX)

**S2 Table. Results from age-adjusted cox regression model where SNP status and warfarin use in evaluated on overall death, cancer-specific death and prostate cancer (PCa)-specific death.** Analysis was limited to screening cohort. Non-warfarin users were used as a reference group. Data is presented as hazard ratios with 95% confidence intervals (HR (95% CI)).
(XLSX)

**S3 Table. Risk of cardiovascular disease (CVD) death by SNP status in warfarin users compared to non-users.**
(XLSX)

**S4 Table.** Categorization of *CYP2C9* by rs1057910 and rs1799853 expression.
(XLSX)

## Author contributions

**Conceptualization:** Teemu J. Murtola, Kimmo Taari, Teuvo L.J. Tammela, Johanna Schleutker, Anssi Auvinen, Csilla Sipeky.

**Data curation:** Teemu J. Murtola, Kaisa M. Skantsi, Kirsi Talala, Johanna Schleutker.

**Formal analysis:** Teemu J. Murtola, Aino Siltari, Csilla Sipeky.

**Funding acquisition:** Teemu J. Murtola, Anssi Auvinen.

**Investigation:** Teemu J. Murtola, Aino Siltari, Kimmo Taari, Teuvo L.J. Tammela, Johanna Schleutker, Anssi Auvinen, Csilla Sipeky.

**Methodology:** Teemu J. Murtola, Aino Siltari, Johanna Schleutker, Anssi Auvinen.

**Project administration:** Kirsi Talala.

**Resources:** Kirsi Talala.

**Supervision:** Teemu J. Murtola, Anssi Auvinen.

**Visualization:** Aino Siltari.

**Writing – original draft:** Teemu J. Murtola, Kaisa M. Skantsi, Aino Siltari.

**Writing – review & editing:** Teemu J. Murtola, Aino Siltari, Kirsi Talala, Kimmo Taari, Teuvo L.J. Tammela, Johanna Schleutker, Anssi Auvinen, Csilla Sipeky.

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
