## [Decision Letter · Decision Letter 0]

13 Jan 2025

Dear Dr. Murtola,

- language problems should be removed,

- presentation problems must be fixed,

- details of the cohort description should be made clear,

- information about the statistical post-hoc analysis and about data distributions would increase the value of the manuscript,

- internet links used in the references list should be verified and updated ([1] - link is not actual; are Authors sure that such unstable and not reviewed sources should be used as references?).

We look forward to receiving your revised manuscript.

Kind regards,

Maciej Huk, Ph.D.

Academic Editor

PLOS ONE

Journal Requirements:

“FinRSPC has been supported by grants from Academy of Finland (grant #260931), Cancer Foundation Finland sr, and Expert Responsibility Area of the Pirkanmaa Hospital District (grant #9V065) (A.A). This study was supported by grant from Cancer Foundation Finland (grant #220055) (T.M). The funders had no role in the study design, data collection and analysis, decision to publish, or preparation of the manuscript.”

4. In the online submission form, you indicated that your data is available only on request from a third party. Please note that your Data Availability Statement is currently missing [the name of the third party contact or institution / contact details for the third party, such as an email address or a link to where data requests can be made]. Please update your statement with the missing information.

“FinRSPC has been supported by grants from Academy of Finland (grant #260931), Cancer Foundation Finland sr, and Expert Responsibility Area of the Pirkanmaa Hospital District (grant #9V065) (A.A). This study was supported by grant from Cancer Foundation Finland (grant #220055) (T.M). The funders had no role in the study design, data collection and analysis, decision to publish, or preparation of the manuscript.”

“FinRSPC has been supported by grants from Academy of Finland (grant #260931), Cancer Foundation Finland sr, and Expert Responsibility Area of the Pirkanmaa Hospital District (grant #9V065) (A.A). This study was supported by grant from Cancer Foundation Finland (grant #220055) (T.M). The funders had no role in the study design, data collection and analysis, decision to publish, or preparation of the manuscript.”

Reviewers' comments:

Reviewer's Responses to Questions

**Comments to the Author**

1. Is the manuscript technically sound, and do the data support the conclusions?

Reviewer #1: Partly

Reviewer #2: Yes

2. Has the statistical analysis been performed appropriately and rigorously?

Reviewer #1: No

Reviewer #2: Yes

3. Have the authors made all data underlying the findings in their manuscript fully available?

Reviewer #1: No

Reviewer #2: No

4. Is the manuscript presented in an intelligible fashion and written in standard English?

Reviewer #1: Yes

Reviewer #2: Yes

Reviewer #1: This text describes the population characteristics of CYP2C9 and VKORC1 genotypes in two study cohorts. The content is clear and logically sound, but there are still some shortcomings and suggestions for improvement:

There is a lack of logical connection between the discussion of prostate cancer risk factors and the coagulation cascade between lines 52 and 53 on page 3. The sections discussing the in vitro studies of warfarin and the current epidemiological situation between lines 60-65 need to be refined for clarity.

Although the methods section mentions that data is sourced from the national cancer registry and hospital records, there is a lack of discussion regarding data quality assessment when describing how to "supplement" cancer diagnosis information. The specific values of the International Normalized Ratio (INR) are not provided, which prevents the classification of patients into good responders and poor responders for analysis. INR is an important indicator for evaluating the anticoagulant effect of warfarin, and the absence of this data may affect the in-depth analysis of warfarin dosage adjustment and efficacy assessment.

The first part of the results mentions differences between genotypes and clinical standards (such as Gleason scores, tumor stages, etc.), but it does not discuss the clinical significance or potential reasons for these results. Please supplement with relevant research evidence.

The results list the hazard ratios (HR) and confidence intervals (CI) for different genotypes, but the clinical significance and biological mechanisms of these results are not discussed, especially regarding why certain genotypes are associated with mortality risk while others are not.

Reviewer #2: ==Language problems==

1. pp 3-4, lines 67-68: Missing article, recommended change: ‘Warfarin targets Vitamin K epOxide Reductase Complex subunit 1 (VKORC1) by inhibiting its activity, an enzyme involved in the vitamin K cycle.’

2. p 5, line 109: Recommended change for improving the readability: ‘The data were accessed for research purposes between January 1, 2022, and October 31, 2024.’

3. p 7, line 140: ‘Only 0.9% of subjects were carrier’ => ‘Only 0.9% of subjects were carriers’

4. Caption of Figure 2: ‘warfaring-metabolizing’ => ‘warfarin-metabolizing’;

5. Caption of Table 4: ‘Data is expressed as hazard ration’ => ‘Data is expressed as hazard ratios’;

==Presentation problems==

1. p 9, lines 188-193: Table 1: Lacking indication that the numbers in parentheses represent the percentages for the row ‘n of men’. Some numbers in the table don’t sum up to the values declared in the caption, e. g. n of men in the screening cohort. Moreover, it is unclear what was intended to be shown regarding the data stratification by SNP type. The description in the caption suggests that the stratification is held in the table, whereas the table itself presents the summed data. It’s recommended to harmonize the description and the tabular data in this context.

2. Figure 1: It is recommended to improve the readability of the cause of death mapping on the graph either by enlarging the font of the geometric figures or by adding color as an auxiliary mapping.

3. Figure 2: The recommendations are similar to those for Figure 1. Apart from that, the figure does not clearly reflect the stratification declared in the caption. It can be deduced from the caption that the second, third, and fourth graphs, as viewed from top to bottom, pertain to warfarin users referenced to the non-users but this information may be overlooked by the reader when scanning the article. Thus, it might be helpful to include this reference in small font individually below the captions of each graph. Additionally, what about the division into two cohorts? Consistently with the caption of Figure 1, I would expect the separate presentation of the results for each of these cohorts. Are they summed up in the Figure 2? If so, it is recommended to indicate this in the caption.

4. Table 2: Adding information to the caption that ‘Ref’ stands for ‘reference group’ would improve the readability of the table. Similarly to the Table 1, it is also recommended to check whether the summations of cases are correct within the cohorts and genotypes.

==Content-related questions and recommendations==

1. p 6, lines 113-114: ‘The study cohort was linked to the national prescription database of the Social Insurance Institution of Finland …’. It is recommended to make this description more precise regarding the cohort or cohorts it refers to. Does this description refer to the screening cohort, the hospital cohort or both? Another fragment which occurs in the lines 143-146 suggests that only screening cohort is concerned but it would be with benefit to readability to mark this information explicitly.

2. p 8, lines 164-165: ‘Interaction was statistically tested including interaction term between warfarin use and SNP genotype into Cox regression model’. Please provide the name of the test used for the evaluation of the interaction term: is it Wald test, Likelihood Ratio Test or other?

3. p 8, lines 177-178: ‘otherwise the distributions of wild type and heterozygous SNP carriers were similar in the two study cohorts’. These frequencies are indeed similar at a glance, but it would be advisable to test the independence of the aforementioned distributions by e.g. Chi-squared test for greater precision and provide the calculated p-value. This remark is more general and refers to other fragments where distributions/frequencies are compared.

RECOMMENDATION: Minor revision

**Do you want your identity to be public for this peer review?** For information about this choice, including consent withdrawal, please see our Privacy Policy

Reviewer #1: No

Reviewer #2: No

---

## [Author Response · Author response to Decision Letter 1]

13 Mar 2025

RESPONSES TO EDITORIAL OFFICE REQUIREMENTS:

Journal Requirements:

Requirement 1. Please ensure that your manuscript meets PLOS ONE's style requirements, including those for file naming. The PLOS ONE style templates can be found at

Response: Thank you for providing the links for formatting samples. We have now updated our manuscript according to these instructions.

Requirement 2. Thank you for stating in your Funding Statement:

“FinRSPC has been supported by grants from Academy of Finland (grant #260931), Cancer Foundation Finland sr, and Expert Responsibility Area of the Pirkanmaa Hospital District (grant #9V065) (A.A). This study was supported by grant from Cancer Foundation Finland (grant #220055) (T.M). The funders had no role in the study design, data collection and analysis, decision to publish, or preparation of the manuscript.”

Response: We have now updated the Funding Statement, removed it from manuscript file and added the updated version to a part of our cover letter of revised manuscript.

Requirement 3. We note that you have indicated that there are restrictions to data sharing for this study. For studies involving human research participant data or other sensitive data, we encourage authors to share de-identified or anonymized data. However, when data cannot be publicly shared for ethical reasons, we allow authors to make their data sets available upon request. For information on unacceptable data access restrictions, please see http://journals.plos.org/plosone/s/data-availability#loc-unacceptable-data-access-restrictions.

Response: We have now updated our Data Availability statement in the manuscript accordingly: There are legal restriction to share the data openly as our data is considered sensitive personal health data. Data access can be applied from Finnish Social and Health Data Authority, Findata (contact information: https://findata.fi/en/ and info@findata.fi).

Requirement 4. In the online submission form, you indicated that your data is available only on request from a third party. Please note that your Data Availability Statement is currently missing [the name of the third-party contact or institution / contact details for the third party, such as an email address or a link to where data requests can be made]. Please update your statement with the missing information.

Response: We have now added this information to our data availability statement (see above).

“FinRSPC has been supported by grants from Academy of Finland (grant #260931), Cancer Foundation Finland sr, and Expert Responsibility Area of the Pirkanmaa Hospital District (grant #9V065) (A.A). This study was supported by grant from Cancer Foundation Finland (grant #220055) (T.M). The funders had no role in the study design, data collection and analysis, decision to publish, or preparation of the manuscript.”

“FinRSPC has been supported by grants from Academy of Finland (grant #260931), Cancer Foundation Finland sr, and Expert Responsibility Area of the Pirkanmaa Hospital District (grant #9V065) (A.A). This study was supported by grant from Cancer Foundation Finland (grant #220055) (T.M). The funders had no role in the study design, data collection and analysis, decision to publish, or preparation of the manuscript.”

Response: We have now removed Funding Statement from the manuscript file and added updated statement as a part of our cover letter.

Requirement 6. Please review your reference list to ensure that it is complete and correct. If you have cited papers that have been retracted, please include the rationale for doing so in the manuscript text, or remove these references and replace them with relevant current references. Any changes to the reference list should be mentioned in the rebuttal letter that accompanies your revised manuscript. If you need to cite a retracted article, indicate the article’s retracted status in the References list and also include a citation and full reference for the retraction notice.

Response: We are thankful for your suggestions and have now gone through our references. Unfortunately, we did not find any retracted publication from the reference list (based on screening using PubMed). If there indeed is one, we can replace it with another one, however, editorial office is asked to point out which reference is under this concern. We found one publication with erratum and information of this is now added to the reference list [24].

RESPONSES TO REVIWERS COMMENTS:

Reviewer #1: This text describes the population characteristics of CYP2C9 and VKORC1 genotypes in two study cohorts. The content is clear and logically sound, but there are still some shortcomings and suggestions for improvement:

Comment: There is a lack of logical connection between the discussion of prostate cancer risk factors and the coagulation cascade between lines 52 and 53 on page 3. The sections discussing the in vitro studies of warfarin and the current epidemiological situation between lines 60-65 need to be refined for clarity.

Response: Thank you for these comments.

We have expanded our introduction on how coagulation cascade may aid cancer cells and modified the text for greater clarity. Introduction, 2nd and 3rd paragraph:

“The coagulation cascade has been proposed to play a role in hematogenous spread of cancer and formation of metastases. Platelets adhere to circulating cancer cells, protecting them from the immune system [5-7]. Platelet RNA is altered in cancer patients, and platelets could serve as potential cancer biomarker [8]. Therefore, it would be logical to assume that anticoagulants, which affect platelet function, would decrease metastatic spread by reducing platelet protection of circulating cancer cells.

Previous pre-clinical studies have reported that widely used anticoagulant warfarin may protect against tumor cell spreading by mechanism related on modulation of anticoagulation cascade rather than a direct impact on cancer cells [9-11]. HFor PCa, warfarin use has been associated with decreased risk in one epidemiological study [12], but not all studies agree [13]. Furthermore in epidemiological studies of cancer mortality, the risk of cancer death is increased among warfarin users compared to non-users [14-16]. “

Comment: Although the methods section mentions that data is sourced from the national cancer registry and hospital records, there is a lack of discussion regarding data quality assessment when describing how to "supplement" cancer diagnosis information. The specific values of the International Normalized Ratio (INR) are not provided, which prevents the classification of patients into good responders and poor responders for analysis. INR is an important indicator for evaluating the anticoagulant effect of warfarin, and the absence of this data may affect the in-depth analysis of warfarin dosage adjustment and efficacy assessment.

Response: We have now added information of the data quality of Finnish register data to the material and method section (page 5, lines 98-100 and 105-108). We agree that INR values are important indicator for evaluation of anticoagulation efficacy of warfarin. Unfortunately, we had no access to the INR values, and could not use them to analyse response to warfarin therapy. We mention this point to our Discussion on study limitations, page 20, lines 14-15:

“However in these data sets, we did not have access on INR values, thus, these analyses need to be done using other data sets.”

Comment: The first part of the results mentions differences between genotypes and clinical standards (such as Gleason scores, tumor stages, etc.), but it does not discuss the clinical significance or potential reasons for these results. Please supplement with relevant research evidence.

Response: We have now added information why cases are more severe in the Hospital cohort than in the Screening cohort (page 10, lines 201-207):

“Differences in the clinical parameters of PCa between the cohorts can largely be attributed to the nature of the cohorts themselves. The Hospital cohort comprises patients with symptom-detected PCa,. Consequently, this cohort includes higher proportion of aggressive and advanced cases, with fewer low-grade PCa cases. In contrast, the Screening cohort consists of PCa cases identified through population-level PSA screening program. Most screening-detected PCa cases were asymptomatic and involved low-grade tumors.”

Comment: The results list the hazard ratios (HR) and confidence intervals (CI) for different genotypes, but the clinical significance and biological mechanisms of these results are not discussed, especially regarding why certain genotypes are associated with mortality risk while others are not.

Response:

All tested SNP genotypes affect warfarin metabolism. Observed differences between them are hypothesis-generating prompting further study to elucidate the mechanistic differences. Currently there is too little evidence to discuss why them.

Instead, we discuss generally why warfarin metabolism might affect prostate cancer outcomes and what it would mean clinically.

Reviewer #2:

Comment:

==Language problems==

1. pp 3-4, lines 67-68: Missing article, recommended change: ‘Warfarin targets Vitamin K epOxide Reductase Complex subunit 1 (VKORC1) by inhibiting its activity, an enzyme involved in the vitamin K cycle.’

2. p 5, line 109: Recommended change for improving the readability: ‘The data were accessed for research purposes between January 1, 2022, and October 31, 2024.’

3. p 7, line 140: ‘Only 0.9% of subjects were carrier’ => ‘Only 0.9% of subjects were carriers’

4. Caption of Figure 2: ‘warfaring-metabolizing’ => ‘warfarin-metabolizing’;

5. Caption of Table 4: ‘Data is expressed as hazard ration’ => ‘Data is expressed as hazard ratios’;

Response: We are thankful for careful proofreading of our manuscript. All above mention language problems have been corrected as requested.

==Presentation problems==

Comment: 1. p 9, lines 188-193: Table 1: Lacking indication that the numbers in parentheses represent the percentages for the row ‘n of men’. Some numbers in the table don’t sum up to the values declared in the caption, e. g. n of men in the screening cohort. Moreover, it is unclear what was intended to be shown regarding the data stratification by SNP type. The description in the caption suggests that the stratification is held in the table, whereas the table itself presents the summed data. It’s recommended to harmonize the description and the tabular data in this context.

Response: Thank you for these comments. We have now added missing information to the table and clarified how stratification was done. We have also clarified information of the values in the caption and now numbers are the same in the caption and in the table.

Comment: 2. Figure 1: It is recommended to improve the readability of the cause of death mapping on the graph either by enlarging the font of the geometric figures or by adding color as an auxiliary mapping.

Response: Thank you for this notion. We have revised the Figure to improve readability

Comment: 3. Figure 2: The recommendations are similar to those for Figure 1. Apart from that, the figure does not clearly reflect the stratification declared in the caption. It can be deduced from the caption that the second, third, and fourth graphs, as viewed from top to bottom, pertain to warfarin users referenced to the non-users but this information may be overlooked by the reader when scanning the article. Thus, it might be helpful to include this reference in small font individually below the captions of each graph. Additionally, what about the division into two cohorts? Consistently with the caption of Figure 1, I would expect the separate presentation of the results for each of these cohorts. Are they summed up in the Figure 2? If so, it is recommended to indicate this in the caption.

Response: We have now added reference group and title for the whole figure to the Fig 2. Hopefully this clarifies the message of the image. Only Screening cohort was used to evaluate how warfarin user status impact on the hazard ratios. We have now clarified that in the title and caption.

Comment: 4. Table 2: Adding information to the caption that ‘Ref’ stands for ‘reference group’ would improve the readability of the table. Similarly to the Table 1, it is also recommended to check whether the summations of cases are correct within the cohorts and genotypes.

Response: We have now added information of reference group into the titles. We have gone through all our summations of the cases and corrected the information to the text and tables. Altogether, we have 810 and 1527 cases in each cohort, however, information of CYP2C9 SNP distribution was available only for 790 and 1437 cases, thus, analyses where these SNPs were evaluated was limited to lower number of PCa cases. This explains the differences in the summed values in the tables. We have now clarified this in the text (page 7, lines 146-150) and titles of the tables.

==Content-related questions and recommendations==

Comment: 1. p 6, lines 113-114: ‘The study cohort was linked to the national prescription database of the Social Insurance Institution of Finland …’. It is recommended to make this description more precise regarding the cohort or cohorts it refers to. Does this description refer to the screening cohort, the hospital cohort or both? Another fragment which occurs in the lines 143-146 suggests that only screening cohort is concerned but it wo

---

## [Decision Letter · Decision Letter 1]

10 Apr 2025

Dear Dr. Murtola,

language and presentation problems should be removed,discussion of results should be extended,description of experimental setup (statistical analysis software) need minor updates,clinical utility of the study should be clearly expressed.

We look forward to receiving your revised manuscript.

Kind regards,

Maciej Huk, Ph.D.

Academic Editor

PLOS ONE

Journal Requirements:

Reviewers' comments:

Reviewer's Responses to Questions

**Comments to the Author**

Reviewer #2: All comments have been addressed

Reviewer #3: All comments have been addressed

Reviewer #4: All comments have been addressed

Reviewer #5: (No Response)

Reviewer #6: All comments have been addressed

2. Is the manuscript technically sound, and do the data support the conclusions?

Reviewer #2: Yes

Reviewer #3: Yes

Reviewer #4: Yes

Reviewer #5: (No Response)

Reviewer #6: Yes

3. Has the statistical analysis been performed appropriately and rigorously?

Reviewer #2: Yes

Reviewer #3: Yes

Reviewer #4: Yes

Reviewer #5: (No Response)

Reviewer #6: Yes

4. Have the authors made all data underlying the findings in their manuscript fully available?

Reviewer #2: No

Reviewer #3: Yes

Reviewer #4: Yes

Reviewer #5: (No Response)

Reviewer #6: Yes

5. Is the manuscript presented in an intelligible fashion and written in standard English?

Reviewer #2: Yes

Reviewer #3: Yes

Reviewer #4: Yes

Reviewer #5: (No Response)

Reviewer #6: No

Reviewer #2: The authors have addressed the recommendations outlined in my review and implemented the suggested improvements. Therefore, I recommend the acceptance of the submitted paper.

Reviewer #3: All comments have been addressed and I have no further comments to discuss with authors or editors on this manuscript.

Reviewer #4: 1. There are some grammatical, alignments and typographical errors are noted in the manuscript and it should be thoroughly checked and corrected throughout the manuscript.

2. Check the abbreviations throughout the manuscript and introduce the abbreviation when the full word appears the first time in the abstract and the remaining for the text and then use only the abbreviation (For example, Prostate cancer (PCa), CVD, etc.,). Make a word abbreviated in the article that is repeated at least three times in the text, not all words to be abbreviated.

3. In introduction, the authors may cite prevalence or incidence data “Prostate cancer” and it should be in either 2024 or 2025.

4. When referring to SPSS versions beginning from 19, authors should cite ‘IBM SPSS Statistics for Windows, version 21 (IBM Corp., Armonk, N.Y., USA)'.

5. The version of the software used for statistical investigation (Stata) in the present investigation should be given under the heading “Statistical analysis”.

6. The authors may improve the discussion of their results by focusing on the present findings and introducing data from other authors who also worked with the same or other studies with recent references since it is lack of sufficient references.

Reviewer #5: The manuscript titled "SNP genotypes in CYP2C9 and VKORC1 genes do not affect prostate cancer or cancer mortality among warfarin users in Finnish prostate cancer patients" is well-conducted. I read the previous comments from the reviewers. I have some additional comments for improvement:

1- Define abbreviations in their first use and make sure that abbreviated forms are used after the definition.

2- In the discussion, the limitations and strengths should be in a separate heading, just before the conclusions.

3- The discussion section is not comprehensive. Add more data from previous meta-analysis and compare them with your findings.

4- Mention the clinical utility of your findings in the discussion.

5- I found some typos and grammatical errors. A native review is warranted.

Reviewer #6: I think the paper is sound and all comments have been addressed .I think in general the paper is solid and should be published.Besides the results are very usefull for the scientific community involved in this subject

**Do you want your identity to be public for this peer review?** For information about this choice, including consent withdrawal, please see our Privacy Policy

Reviewer #2: No

Reviewer #3: No

Reviewer #4: **Yes: ** Dr. A. Vijaya Anand

Reviewer #5: No

Reviewer #6: No

---

## [Author Response · Author response to Decision Letter 2]

25 Apr 2025

The coagulation cascade is thought to contribute to cancer progression. Although in vitro studies suggest that anticoagulants, such as warfarin, might reduce cancer progression, epidemiological data indicate that warfarin users may have a higher risk of cancer mortality. However, single nucleotide polymorphisms (SNPs) that influence warfarin dosing might affect this association. We investigated the risk associations between warfarin use and prostate cancer (PCa) survival, considering the SNP genotypes of CYP2C9 and VKORC1, which are known to impact both warfarin pharmacokinetics and pharmacodynamics, resulting in lower warfarin dose requirement. We genotyped 2,246 Finnish men with PCa from two different cohorts for SNPs rs1057910, rs1799853, and rs9923231. Genotyping was done using a custom Illumina iSelect genotyping array (iCOGs). Using Cox regression models, we calculated hazard ratios (HRs) and 95% confidence intervals (CI) for the risk of overall death, cancer deaths overall, and PCa-specific death after PCa diagnosis based on SNP genotypes. Data on warfarin purchases was obtained from a national registry. Our findings revealed that the SNPs did not alter the risk of cancer or PCa death in either cohort, nor did they modify the risk among warfarin users. However, overall mortality was higher among warfarin users compared to non-users, particularly in carriers of all three SNPs. Even though the increased mortality is likely due to confounding by indication, warfarin use may increase overall mortality especially in men with lower warfarin dose requirements due to SNP carrier status. However, we need further studies with larger populations to confirm these findings.

---

## [Decision Letter · Decision Letter 2]

26 May 2025

Dear Dr. Murtola,

**In particular:**

**grammatical, alignment and typographical errors need to be removed,****all abbreviations should be defined before their first use.**

We look forward to receiving your revised manuscript.

Kind regards,

Maciej Huk, Ph.D.

Academic Editor

PLOS ONE

**Journal Requirements:**

Reviewers' comments:

Reviewer's Responses to Questions

**Comments to the Author**

Reviewer #4: All comments have been addressed

Reviewer #5: All comments have been addressed

2. Is the manuscript technically sound, and do the data support the conclusions?

Reviewer #4: Yes

Reviewer #5: (No Response)

3. Has the statistical analysis been performed appropriately and rigorously?

Reviewer #4: Yes

Reviewer #5: (No Response)

4. Have the authors made all data underlying the findings in their manuscript fully available?

Reviewer #4: Yes

Reviewer #5: (No Response)

5. Is the manuscript presented in an intelligible fashion and written in standard English?

Reviewer #4: Yes

Reviewer #5: (No Response)

**Reviewer #4:**  1. There are some grammatical, alignments and typographical errors are noted in the manuscript and it should be thoroughly checked and corrected throughout the manuscript. For example,

• in line number 70, the words “causal” may be as “the causal”;

• in line number 92, “Tampere” as “the Tampere”;

• in line number 151, “Information of” as “Information on”;

• in line number 168, “for death” as “death”;

• in line number 175, “Interaction” as “The interaction”;

• in line number 199, “Gleason” as “a Gleason”;

• in line number 211, “diagnosis” as “a diagnosis”;

• in line number 217, “Similarly,no” as “Similarly, no”;

• in line number 221, “f PCa” as “of PCa”;

• all over the manuscript, “as hazard” as “as a hazard”;

• in line number 6, “anddid” as “and did”;

• in line number 24, “wecannot dretermine” as “we can not determine”;

• in line number 2, “factors..” as “factors.”;

• in line number 17, “International” as “the International”;

• in line number 22, “history” as “a history”;

• in line number 17, “groupss is” as “groups are”;

• in line number 8, “larger” as “a larger”.

2. This suggestion is not carried out properly and it should be rectified. Check the abbreviations throughout the manuscript and introduce the abbreviation when the full word appears the first time in the abstract and the remaining for the text and then use only the abbreviation (For example, Prostate cancer (PCa) and this types of corrections need to be checked all other abbreviations used in the manuscript.

**Reviewer #5: ** (No Response)

**Do you want your identity to be public for this peer review?** For information about this choice, including consent withdrawal, please see our Privacy Policy

Reviewer #4: **Yes: ** Dr. A. Vijaya Anand

Reviewer #5: No

---

## [Author Response · Author response to Decision Letter 3]

9 Jul 2025

Responses to reviewers’ comments

July 9th, 2025

We are thankful for the comments and made modifications to our manuscript based on them. All changes are marked at the revised version of the manuscript. Please, below you can find our responses to reviewers’ comments.

Review Comments to the Author:

Comment: Reviewer #2: The authors have addressed the recommendations outlined in my review and implemented the suggested improvements. Therefore, I recommend the acceptance of the submitted paper.

Response: Thank you for your positive feedback.

Comment: Reviewer #3: All comments have been addressed and I have no further comments to discuss with authors or editors on this manuscript.

Response: Thank you for your positive feedback.

Reviewer #4:

Comment: 1. There are some grammatical, alignments and typographical errors are noted in the manuscript and it should be thoroughly checked and corrected throughout the manuscript.

Response: We have now checked and corrected the text from grammatical, alignments and typographical errors.

Comment: 2. Check the abbreviations throughout the manuscript and introduce the abbreviation when the full word appears the first time in the abstract and the remaining for the text and then use only the abbreviation (For example, Prostate cancer (PCa), CVD, etc.,). Make a word abbreviated in the article that is repeated at least three times in the text, not all words to be abbreviated.

Response: We have now gone through the manuscript and clarify the use of abbreviations throughout the manuscript.

Comment: 3. In introduction, the authors may cite prevalence or incidence data “Prostate cancer” and it should be in either 2024 or 2025.

Response: Unfortunately, the latest statistics of prevalence and incidence rates are from the year 2022 (Bray F et al. Global cancer statistics 2022: GLOBOCAN estimates of incidence and mortality worldwide for 36 cancers in 185 countries. CA Cancer J Clin. 2024 May-Jun;74(3):229-263.). We have cited the publication using data from 2022 which estimate the current numbers. Related on prostate cancer incidence data, in our opinion, the reference used represents the current situation.

Comment: 4. When referring to SPSS versions beginning from 19, authors should cite ‘IBM SPSS Statistics for Windows, version 21 (IBM Corp., Armonk, N.Y., USA)'.

Response: As suggested, we have now added more detailed information on the statistical programs we used (page 9, lines:189-190).

Comment: 5. The version of the software used for statistical investigation (Stata) in the present investigation should be given under the heading “Statistical analysis”.

Response: We have added the requested information under the heading of Statistical analysis (page 9, line:190).

Comment: 6. The authors may improve the discussion of their results by focusing on the present findings and introducing data from other authors who also worked with the same or other studies with recent references since it is lack of sufficient references.

Response: We have now added more comprehensive discussion on reflection of our findings on previous knowledge of the topic (page: 17, lines:11-20):

“To our knowledge, this is the first study to evaluate the association of SNPs rs1057910, rs1799853, and rs9923231 with PCa progression. However, SNP rs1057910 has previously been associated with an increased risk of developing colorectal carcinoma [33]. We did not assess whether the investigated SNPs are associated with the risk of developing PCa, therefore, this should be explored using other cohorts that include genotyping data from healthy controls as well. In PCa, VKORC1 gene expression has been studied in prostate tissue, and it was found to be higher in benign tissue compared to PCa tissue. Interestingly, SNP rs2359612, which also influences warfarin dosing, was linked to PCa risk [34]. Notably, based on a genome-wide association study (GWAS), none of the investigated SNPs are among the 451 independent variants reported to be associated with PCa risk [35].”

Reviewer #5: The manuscript titled "SNP genotypes in CYP2C9 and VKORC1 genes do not affect prostate cancer or cancer mortality among warfarin users in Finnish prostate cancer patients" is well-conducted. I read the previous comments from the reviewers. I have some additional comments for improvement:

Comment: 1- Define abbreviations in their first use and make sure that abbreviated forms are used after the definition.

Response: We have now gone through the manuscript and clarify the use of abbreviations throughout the manuscript.

Comment: 2- In the discussion, the limitations and strengths should be in a separate heading, just before the conclusions.

Response: We have added separate headings as suggested (page 19).

Comment: 3- The discussion section is not comprehensive. Add more data from previous meta-analysis and compare them with your findings.

Response: We have now added more comprehensive discussion on reflection of our findings on previous knowledge on the topic. We have also compared our findings to a recent GWAS meta-analysis of gene variants associated with prostate cancer risk (page: 17, lines:11-20).

“To our knowledge, this is the first study to evaluate the association of SNPs rs1057910, rs1799853, and rs9923231 with PCa progression. However, SNP rs1057910 has previously been associated with an increased risk of developing colorectal carcinoma [33]. We did not assess whether the investigated SNPs are associated with the risk of developing PCa, therefore, this should be explored using other cohorts that include genotyping data from healthy controls as well. In PCa, VKORC1 gene expression has been studied in prostate tissue, and it was found to be higher in benign tissue compared to PCa tissue. Interestingly, SNP rs2359612, which also influences warfarin dosing, was linked to PCa risk [34]. Notably, based on a genome-wide association study (GWAS), none of the investigated SNPs are among the 451 independent variants reported to be associated with PCa risk [35].”

Comment: 4- Mention the clinical utility of your findings in the discussion.

Response: Thank you for pointing this out. We have now added the sentence clarifying this point in the beginning of the discussion (page: 17, lines: 10-12).

“Thus, this indicates that the genetic factors influencing warfarin dosing do not impact the potential benefits cancer patients may receive from using warfarin.”

Comment: 5- I found some typos and grammatical errors. A native review is warranted.

Response: We have now checked and corrected the text from grammatical, alignments and typographical errors.

Comment: Reviewer #6: I think the paper is sound and all comments have been addressed. I think in general the paper is solid and should be published. Besides the results are very usefull for the scientific community involved in this subject.

Response: Thank you for your positive feedback.

---

## [Decision Letter · Decision Letter 3]

17 Jul 2025

SNP genotypes in CYP2C9 and VKORC1 genes do not affect prostate cancer or cancer mortality among warfarin users in Finnish prostate cancer patients

PONE-D-24-50316R3

Dear Dr. Murtola,

We’re pleased to inform you that your manuscript has been judged scientifically suitable for publication and will be formally accepted for publication once it meets all outstanding technical requirements.

Kind regards,

Maciej Huk, Ph.D.

Academic Editor

PLOS ONE

Additional Editor Comments (optional):

Reviewers' comments:

Reviewer's Responses to Questions

**Comments to the Author**

Reviewer #4: All comments have been addressed

2. Is the manuscript technically sound, and do the data support the conclusions?

Reviewer #4: Yes

3. Has the statistical analysis been performed appropriately and rigorously?

Reviewer #4: Yes

4. Have the authors made all data underlying the findings in their manuscript fully available?

Reviewer #4: Yes

5. Is the manuscript presented in an intelligible fashion and written in standard English?

Reviewer #4: Yes

Reviewer #4: 1. This suggestion is not carried out properly and it should be rectified. Check the abbreviations throughout the manuscript and introduce the abbreviation when the full word appears the first time in the abstract and the remaining for the text and then use only the abbreviation (For example, in line number 164, “prostate cancer” should be replaced as “PCa” and this types of corrections need to be checked all other abbreviations used in the manuscript.

**Do you want your identity to be public for this peer review?** For information about this choice, including consent withdrawal, please see our Privacy Policy

Reviewer #4: **Yes: ** Dr. A. Vijaya Anand

---

## [Editor Report · Acceptance letter]

PONE-D-24-50316R3

PLOS ONE

Dear Dr. Murtola,

I'm pleased to inform you that your manuscript has been deemed suitable for publication in PLOS ONE. Congratulations! Your manuscript is now being handed over to our production team.

Kind regards,

on behalf of

Dr. Maciej Huk

Academic Editor

PLOS ONE